# Injury patterns of non-fatal accidents related to ice hockey, an analysis of 7 years of admission to a Level-1 Emergency Centre in Switzerland

**Viola Gilardi☉, Spyridon Kotsaris☉, Aristomenis Exadaktylos, Jolanta Klukowska-Rötzler ☉ ***

Department of Emergency Medicine, University Hospital, Bern, Switzerland

☉ These authors contributed equally to this work.
* jolanta.klukowska-roetzler@insel.ch

## Abstract

### Objective

This study was carried out to identify the frequency and types of injuries in adult ice hockey, in order to better understand injury patterns and identify potential areas for injury prevention.

### Methods

We conducted a retrospective database review of acute injuries reported in ice hockey in patients presenting to a Level-1 adult Emergency Centre in Switzerland. Patients between January 1, 2013 and December 31, 2019 and over 16 years of age were identified in our computerised patient database. Each consultation was reviewed to derive information on demographics, playing level and the features of the injury, including location, type, mechanism and consequences. Different age groups were compared, as were amateur and professional players. A statistical analysis was performed.

### Results

A total of 230 patients were identified. The most common diagnoses were fracture (28.3%), contusion/abrasion (23.9%), laceration (12.6%) and concussion (10.4%). The most commonly affected body parts were the face (31.3%), the shoulder/clavicle (13.0%) and the head (12.2%). Most lesions were caused by player-player contact (37.4%), contact with the puck (24.3%) and falls (10.9%). In comparison to the younger cohorts, patients >36 years of age more frequently suffered injuries caused by falls, (p < 0.001) and were less frequently injured by player-player contact (p = 0.01813). In amateur players, significantly more injuries were caused by stick contact (OR 0, 95% CI (0.00–0.83), p = 0.02) and surgery was more rarely performed (OR 2.35, 95% CI 0.98–5.46, p = 0.04).

### Conclusions

Injuries continue to play a major role in ice hockey, especially in the face and due to player-player contact. Future investigations should focus on player-player contact and possible

**Data Availability Statement:** The data for this study comes from the specialist database for patient management in the emergency department. As such, the Ethical Committee in Bern as applied

ethical restrictions on sharing the data publicly. Data Access request can be directed to the following contact: Gesundheits-, Sozial- und Integrationsdirektion Kantonale Ethikkommission für die For-schung (www.be.ch/gsi) Email: info.kek. kapa@be.ch Dorothy Pfiffner Email: dorothy. pfiffner@be.ch. Tel.: +41 31 633 70 77.

**Funding:** The author(s) received no specific funding for this work.

**Competing interests:** The authors have declared that no competing interests exist.

effective preventive measures. Players must be encouraged to employ face protection and to wear a mouth guard at all times.

## Introduction

The present form of men's ice hockey was introduced by British soldiers in Canada in the mid-1800s, and the first rules were laid down in 1879. Nowadays the international ice hockey federation (IIHF) represents 82 national associations and ice hockey is the largest team sport in the Olympic Winter Games [1]. There are currently more than 1,160 ice hockey teams registered in Switzerland, where ice hockey is the national sport with the most supporters after soccer [2]. It is considered to be one of the fastest and most dynamic winter sports, and is characterised by its aggressive nature. It is very rapid and involves frequent physical contact, including collisions between players that regularly lead to injury. When sliding on ice, elite players may reach speeds of up to 48 km/h and the puck can reach a speed of 161 km/h during the game [3]. Not unexpectedly, at the 2010 Winter Olympics, ice hockey was one of the sports with the highest risk of injury and the sport with the highest incidence of injuries caused by contact between athletes [4].

Ice hockey can lead to a broad variety of injuries, from minor complaints (e.g., abrasions, superficial lacerations and contusions) to severe lesions (e.g., fractures, including spine fracture with spinal cord involvement, acute compartmental syndromes, intracranial bleedings and concussions). These lead to more or less invasive therapy and lost time, in which the player is unable to play [5, 6]. Injuries more frequently occur during a game than in practice [3, 7–17] and are more frequent at higher levels [7, 9] and in men's hockey than in women's hockey [11, 17].

In order to enhance the fairness and safety of the game, the International Ice Hockey Federation sets rules for the proper use of equipment and for the punishment of dangerous behaviour. In addition to helmets, male players must wear at least a visor; female players a cage or full-face visor, and players in the under-18 age category a cage [18]. In women's ice hockey, it is not allowed to check an opponent with the body. In men's ice hockey, a check to the head or neck area is subjected to a penalty [18]. However, the accident rate in this sport remains high and there is room for improvement. In addition, individuals who play recreationally may not wear all of the recommended equipment.

Besides the impact on player health, injuries also have socioeconomic implications, due to lost time and the required medical care.

For all of these reasons, ice hockey injuries are a topic of interest for emergency departments. Most studies are carried out on young players and at higher levels of play. Moreover, there have been few European studies on ice hockey injuries in the past two decades. This present study aims to describe patients with acute ice hockey injuries admitted to an adult's Swiss Emergency Department and is intended to improve our understanding of the injury patterns in the hockey population. This should support evidence-based prevention of primary injuries and improve the evaluation and treatment of injuries.

## Material and methods

### Data collection and retrospective survey

The study was conducted at a Level I trauma centre in the capital city of Bern, Switzerland (University Emergency Centre, Inselspital), that treats more than 50,000 cases per year (2019).

The study is based on the demographic and health-related data of patients over 16 years of age. Children under 16 years were not included, as they were admitted to a separate emergency department. Patients who had sustained an acute trauma while playing ice hockey in the period from January 1st 2013 to December 31st 2019 were identified using the search string ("Eishockey") in the diagnosis or medical history field of our computerised patient database E. care (E.care BVBA, ED 2.1.3.0, Turnhout, Belgium). Duplicate case records and outpatient follow-ups were excluded from the analyses. Cases involving injuries to referees and coaches, injuries due to overuse or cases with insufficient information were also excluded. If a general consensus against the use of the patient data was documented, the data were automatically not recorded.

Each consultation was reviewed to derive information on the injury. The demographic variables taken into account are: gender, age, type of referral (self-presentation, ambulance, external doctor, etc.), date of admission, triage priority, case history (accident mechanism), imaging performed, injured body part, diagnosis, as well as information on treatment and discharge (outpatient versus inpatient, admission department, conservative versus surgical treatment). In addition, we looked at the level at which the patients played, i.e., whether they were professionals or not. In case the level was not specified, an additional search was performed using the Elite Prospect database. Elite Prospect is a Swedish database of ice hockey players and a transaction portal. Elite Prospect contains reliable information on teams and players from different countries around the world, including Switzerland, and is therefore a statistical resource for ice hockey [19]. The playing level has been defined as professional for patients belonging to the following teams: National League (NL), Swiss League (SL), Under 20-Elit (U20-Elite) and U17-Elit for men, as well as Swiss Women's League A (SWHL A) for women, because at these levels players usually start to receive payment. Until 2017, the NL was designated as National League A (NLA) and the SL as National League B (NLB). Until 2019, the U20-Elit was designated as Elite Juniors A and the U17-Elite as Elite Novizen [19].

## Analysed data

The regular ice hockey season in Switzerland runs from mid-September/end of September until the end of February/beginning of March. The Playoffs and Playouts follow the regular season. We have placed the midpoint of the regular season at the beginning of December, thus dividing the season into a first part (September-November) and a second part (December-February).

Thanks to the Swiss Emergency Triage Scale (SETS) system based on four levels, it was possible to estimate the severity of the patients' health conditions. The Swiss Emergency Triage Scale (SETS) system incorporates timeliness objectives, where SETS level 1 represents situations of utmost urgency with the need for immediate intervention, SETS level 2 represents potentially life-threatening situations where assessment and treatment should be performed within 20 minutes, SETS level 3 represents situations where assessment and treatment are mandatory within 120 minutes and SETS level 4 represents non-urgent conditions [20].

The patient variables that were compared included sex, age, playing level, leading mechanism of injury, main diagnosis, injured body part and treatment performed. The patients were split by sex and divided into 4 age groups: 16–25 years old, 26–35 years old, 36–45 years old and ≥ 46 years old. Since the older age groups often contained few patients, in order to perform the statistical analysis, the two oldest patient cohorts were included in a single group of patients ≥ 36 years of age. A further division has been made into professional and non-professional players. Injury mechanism has been categorised into the following groups: falls; player-player contact; contact with boards; contact with stick; contact with puck; contact with skate

and other/unknown. In the case that the contact between players caused a fall or a contact with the boards, the injury mechanism was indicated as player-player contact. If the patient was knocked against the boards by the body of an opponent (boarding), this was therefore categorised as player-player contact, but still taken into account for further discussion. As regards the diagnosis, it was initially decided whether patients had suffered a monotrauma, combined trauma or a polytrauma. Polytrauma was defined as simultaneous injuries to several parts of the body where at least one or the combination of several injuries is life-threatening [21, 22]. Subsequently, only the main diagnosis, i.e., the one with the greatest clinical relevance, was taken into account when making a comparison. The main diagnoses were categorised as follows: laceration; fracture; contusion/abrasion; dislocation, sprain/strain (including ligament and meniscal lesions); concussion; intracranial bleeding; dental Injury; eye injury, ear/tympanum lesion and major haematoma. Concussion is defined according to the internationally recognised definition of the Sport Concussion Assessment Tool [23]. Major haematomas were defined as haematomas with a serious clinical presentation that led to hospitalisation for monitoring or for surgery—such as in the case of compartment syndrome. In addition, consideration was given to which part of the body was injured. The treatment performed—surgery versus conservative treatment—was compared for professional and amateur players. A patient with missing data on the treatment performed was excluded from this statistical analysis. Furthermore, patients who came to the emergency room several times during the observed time period were analysed with regard to sex, level of play and type of diagnosis, to determine whether the type of diagnosis had occurred several times over the timeframe.

## Statistical analysis

Descriptive statistics were carried out by calculating frequencies and percentages. The comparisons of injury proportion based on age or playing level was made using the Fisher's exact test, which is also valid for small samples, with 95% confidence intervals (CI). The threshold of significance has been set at a p-value of 0.05. The odds ratio (OR) was calculated whenever two variables were compared. If several variables were compared, such as for different age cohorts, this was not possible and only the p-value was reported. A linear regression model was used to estimate the temporal trend. The software used for the statistical analysis was R (R Foundation for Statistical Computing) version 4.0.2.

## Ethical considerations

The conduct of this descriptive case series was approved by the cantonal (= district) ethics committee in Bern (No 2020–02611). The analysis was carried out with anonymised data. General Consent (Consent for the use of patient data for retrospective research projects) was introduced at the University ED in February 2015. If there is a documented GC refusal, or contradiction to use patient data, these are not recorded. The research ethics committee waived further individual informed consent.

# Results

## Overall considerations and patients' characteristics

From January $1^{st}$ 2013 to December $31^{st}$ 2019, 230 patients aged $\geq$ 16 years presented to our emergency room (ER) after an acute trauma while playing ice hockey. Of these, 97% were men (n = 223) and 3% women (n = 7). The small number of women patients did not allow us to draw any statistically relevant considerations.

The most numerous age category was the youngest (16–25 years), that included 54.8% of cases (n = 126). The number of cases decreased for each subsequent age cohort: 27% for the 26–35 years cohort (n = 62), 11.3% for the 36–45 years cohort (n = 26) and 7% for the ≥46 year old cohort (n = 16). Overall, the youngest patient was 16 and the oldest 64 years old, with a mean age of 27.2 years.

Patients mostly played at a non-professional level, corresponding to 82.2% (n = 182), compared to 17.8% of patients who played at a professional level (n = 41) (Table 1).

Between 2013 and 2019, ER consultations due to ice hockey injuries increased by 33.3% (n = 27 to n = 36). However, this difference was not statistically significant (OR 0.72, 95% CI (0.40–1.27), p = 0.28). There was also no statistically significant increase in the annual number of cases during the 7 years observed (2013–2019) (linear slope 0.54, 95% CI (-1.85–2.92), $R^2$ = 0.06, p = 0.59) (Fig 1).

As expected, the number of cases was significantly higher during the period of the regular season (September-February)—with 83.5% of cases (n = 192)—than outside this period (March-August)—with 16.5% of cases (n = 38) (OR 25.25, 95% CI (15.18–43.05), p < 0.001). No significant difference was observed between the beginning of the regular season (September-November) and its end (December-February), corresponding respectively to 40.0% (n = 92) and 43.5% of cases (n = 100) (OR 0.87, 95% CI (0.59–1.28), p = 0.51).

The vast majority of patients, i.e. 70.4%, presented spontaneously to the emergency room (n = 162); 13.5% were brought by ambulance (n = 31); 6.5% transferred from external clinics (n = 15); 4.8% sent by an external doctor (n = 11); 3.5% transported via REGA (Swiss Air-Rescue) (n = 8) and 1.3% sent to the ER from internal departments of our clinic (n = 3).

8.7% of cases were triaged in the shock room (n = 20). The majority of patients were assigned to the SETS 3 triage level (70%, n = 161), followed by the SETS 2 triage level (21.3%, n = 49). A lower incidence was observed for SETS 1 and SETS 4 levels, both corresponding to a percentage of 4.3% (n = 10). 59.1% of patients presented on the same day as the accident (n = 136); 19.1% on the day after the accident (n = 44) and smaller rates in the following days. For 15.2% of patients, there was no information on the time of admission to the emergency department in relation to the time of injury (n = 35).

As mentioned above, the damages also have socioeconomic consequences. An idea of this is given by the extensive imaging that was carried out, as well as the admissions and operations performed. 43.0% of patients underwent an X-ray (n = 99); 33.9% a computerised tomography (CT) scan (n = 78); 9.6% magnetic resonance imaging (MRI) (n = 22); 5.7% Lodox scanning (n = 13); 11.3% panoramic dental X-ray/orthopantomography (OPT) (n = 26) and 17% sonography (including focused assessment with sonography for trauma, FAST) (n = 39). In 42 patients, corresponding to 18.2%, no imaging was performed.

79.1% of subjects were treated as outpatients (n = 182); 20% were admitted to the hospital (n = 46) and 0.9% were transferred to external clinics (n = 2). Of the hospitalised patients, 45.7% were admitted to the craniomaxillofacial surgery department (n = 21); 28.3% to the department of orthopaedic surgery and traumatology (n = 13); 8.7% to the otorhinolaryngology department (ENT) (n = 4); the same proportion of 6.5% were admitted to the department of hand surgery and intermediate care unit (IMC) (n = 3, n = 3); as well as 2.2% of patients to the department of ophthalmology and intensive care unit (n = 1, n = 1). The mean length of hospital stay was 3.8 days (1–9 days).

The diagnoses that required hospitalisation were as follows: 11 mandibular fractures (mean stay 4.1 days), 6 major haematomas (mean stay 5.8 days), 3 laryngeal traumas (mean stay 2.3 days), 1 open finger fracture (1 day of stay), 4 maxillary fractures (mean stay 2.5 days), 3 dental injuries (mean stay 2 days), 1 bucket-handle meniscal tear (3 days of stay), 1 eye injury (4 days of stay), 3 spinal cord contusions (mean stay 2.3 days), 1 nasal fracture (5 days of stay), 2

**Table 1. Sociodemographic data and overall information of patients with ice hockey injuries presented at Insel-spital University Emergency Centre between 2013 and 2019.**

| Characteristics | Case count (n) | Percentage (%) |
|---|---|---|
| **Sex** | | |
| Male | 223 | 97.0 |
| Female | 7 | 3.0 |
| *Total* | *230* | *100* |
| **Age group (years)** | | |
| 16–25 | 126 | 54.8 |
| 26–35 | 62 | 27.0 |
| 36–45 | 26 | 11.3 |
| ≥ 46 | 16 | 7.0 |
| **Ice hockey playing level** | | |
| Professional level | 41 | 17.8 |
| Non-professional level | 189 | 82.2 |
| **Year** | | |
| 2013 | 27 | 11.7 |
| 2014 | 41 | 17.8 |
| 2015 | 29 | 12.6 |
| 2016 | 31 | 13.5 |
| 2017 | 33 | 14.3 |
| 2018 | 33 | 14.3 |
| 2019 | 36 | 15.7 |
| **Time of occurrence** | | |
| • September-February (regular season) | 192 | 83.5 |
| *- September-November (part 1 of regular season)* | *- 92* | *- 40.0* |
| *- December-February (part 2 of regular season)* | *- 100* | *- 43.5* |
| • March-August (time outside the regular season) | 38 | 16.5 |
| **Type of referral to the hospital** | | |
| Spontaneous presentation | 162 | 70.4 |
| Ambulance | 31 | 13.5 |
| REGA (Swiss Air Rescue) | 8 | 3.5 |
| Family doctor/external doctor | 11 | 4.8 |
| Transfer from external clinic | 15 | 6.5 |
| Transfer from internal clinic | 3 | 1.3 |
| **Urgency** | | |
| **• Shock room** | | |
| *- Shock room* | 20 | 8.7 |
| *- No Shock room* | 210 | 91.3 |
| **• Triage category (SETS)** | | |
| *- 1* | 10 | 4.3 |
| *- 2* | 49 | 21.3 |
| *- 3* | 161 | 70.0 |
| *- 4* | 10 | 4.3 |
| **Admission time** | | |
| directly after trauma | 136 | 59.1 |
| 1 day after trauma | 44 | 19.1 |
| 2 days after trauma | 6 | 2.6 |
| 3 days after trauma | 2 | 0.9 |

*(Continued)*

**Table 1.** (Continued)

| Characteristics | Case count (n) | Percentage (%) |
|---|---|---|
| 4 days after trauma | 0 | 0 |
| 5 days after trauma | 4 | 1.7 |
| 6 days or more after trauma (6–10 days) | 3 | 1.3 |
| Unknown/ no information | 35 | 15.2 |
| *Management* | | |
| Outpatients | 182 | 79.1 |
| Inpatients | 46 | 20 |
| External transfers | 2 | 0.9 |
| *Departments to which patients were admitted* | | |
| Craniomaxillofacial surgery | 21 | 45.7 |
| Orthopaedic surgery and traumatology | 13 | 28.3 |
| Otorhinolaryngology, ENT | 4 | 8.7 |
| Hand Surgery | 3 | 6.5 |
| Ophthalmology | 1 | 2.2 |
| Intermediate care unit, IMC | 3 | 6.5 |
| Intensive care | 1 | 2.2 |
| *Imaging* | | |
| None | 42 | 18.3 |
| X-ray | 99 | 43.0 |
| CT scan | 78 | 33.9 |
| MRI | 18 | 7.8 |
| Lodox scanner | 13 | 5.7 |
| OPT | 26 | 11.3 |
| Sonography | 39 | 17.0 |
| *Treatment* | | |
| Conservative treatment | 189 | 82.2 |
| Surgical treatment | 40 | 17.4 |
| No information | 1 | 0.4 |

lacerations (1 with digital nerve injury, 1 with injury of the facial nerve) (1.5 days of stay), 1 trans-scaphoid perilunate fracture dislocation (9 days of stay), 2 complex maxillofacial fractures (mean stay 3.5 days), 2 distal radius fractures (mean stay 4 days), 1 orbital blowout fracture (5 days of stay), 1 ankle fracture dislocation (5 days of stay), 1 concussion (9 days of stay), 2 intracranial bleedings (mean stay 2.5 days).

85.0% of cases submitted to surgery required a stay in hospital. 82.2% of patients could be treated conservatively (n = 189), while 17.4% underwent surgery (n = 40). For one patient, corresponding to 0.4%, it was not possible to trace the type of treatment performed (Table 1).

Tooth injuries accounted for 7.5% of all surgeries (n = 3), fractures for 40% (n = 28), major haematomas for 7.5% (n = 3), lacerations with injury of more profound structures for 10% (n = 4) and acromioclavicular joint injuries, as well as meniscal tears for 2.5% (n = 1, n = 1).56.9% of fractures were treated conservatively (n = 37), 43.1% by surgery (n = 28). Surgical procedures included 11 mandibular fractures, 4 maxillary fractures, 1 orbital blowout fracture, 2 complex maxillofacial fracture, 1 humerus fracture, 1 open finger fracture, 1 nasal fracture, 1 trans-scaphoid perilunate fracture dislocation, 1 distal forearm fracture, 2 distal radius fractures, 1 thyroid cartilage fracture, 2 ankle fractures (1 ankle fracture dislocation, 1 type Weber B). The hospital departments with the highest proportions of surgeries were the

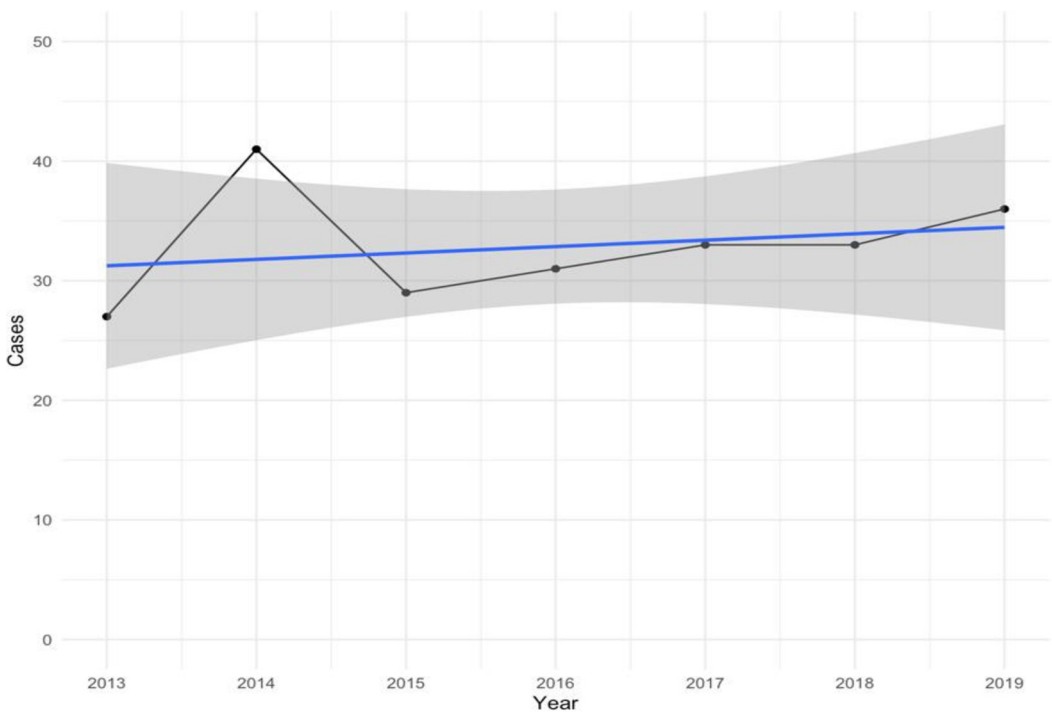

**Fig 1. Annual number of cases with acute ice hockey injuries presented at Inselspital University Emergency Centre between 2013 and 2019.**

craniomaxillofacial surgery and the hand surgery departments. In both cases, 100% of the admitted patients underwent surgical procedures (n = 21 for craniomaxillofacial surgery and n = 3 for hand surgery).

## Overall injury characteristics

90% of patients sustained monotrauma (n = 207), with a combined trauma in 10% (n = 23). No patient reported polytrauma. The most common diagnoses were fracture (28.3%, n = 65), contusion/abrasion (23.9%, n = 55), laceration (12.6%, n = 29) and concussion (10.4%, n = 24). This was followed by sprain and strain (7.4%, n = 17), dislocation (5.2%, n = 12), dental injury (4.3%, n = 10), eye injury (3.5%, n = 8), major haematoma (2.6%, n = 6), ear/tympanum lesion (0.9%, n = 2) and intracranial bleeding (0.9%, n = 2) (Fig 2).

If fractures occurred, 73.8% of these were closed (n = 48), and 26.2% open (n = 17). The site of the fracture was the upper limbs in 44.6% of cases (n = 29); in 41.5% the head and face (n = 27), in 7.7% the lower limbs (n = 5) and in 6.2% the spine (n = 4).

The *anatomical location of injury* was mostly the face (31.3%, n = 72), the shoulder/clavicle (13.0%, n = 30), the head (12.2%, n = 28) and the back/neck/throat (10%, n = 23). Less frequent

| | |
|---|---|
| Intracranial bleedin | 0.9% |
| Ear/Tympanum les | 0.9% |
| Major hematoma | 2.6% |
| Eye injury | 3.5% |
| Dental Injury | 4.3% |
| Dislocation | 5.2% |
| Sprain/strain/ligam | 7.4% |
| Concussion | 10.4% |
| Laceration | 12.6% |
| Contusion/Abrasio | 23.9% |
| Fracture | 28.3% |

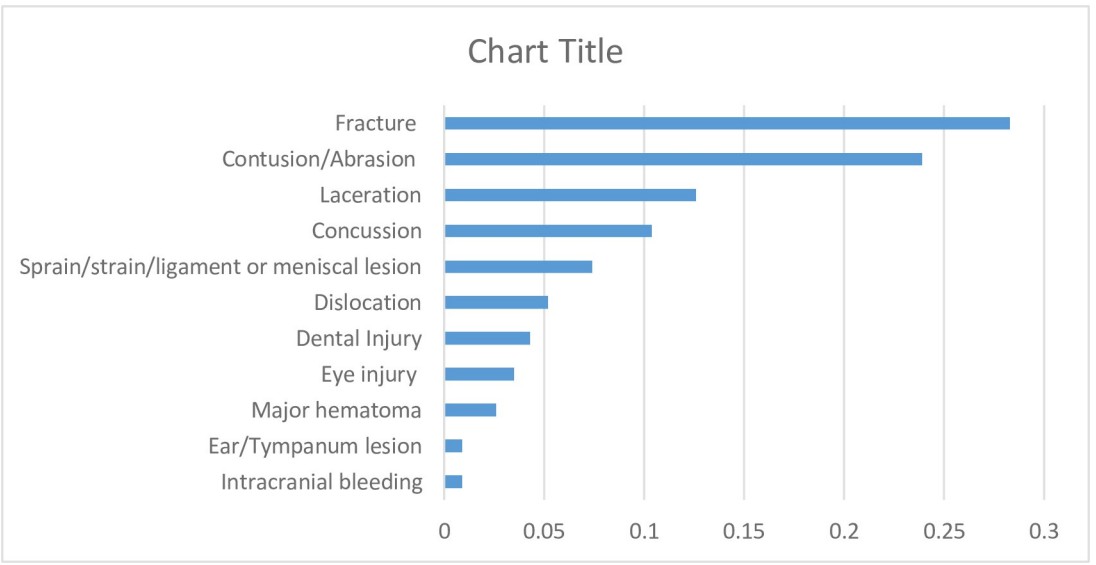

**Fig 2. Frequency of main diagnoses in percent.**

sites of injury included the hand and fingers (7.8%, n = 18), pelvis/hips/thigh (6.5%, n = 15), knee (6.1%, n = 14), wrist (4.3%, n = 10), chest and flank (3.5%, n = 8), ankle (2.6%, n = 6), arm (2.2%, n = 5) and foot/toes (0.4%, n = 1 (Figs 3 and 4).

The types of injury were as follows:

- Shoulder/clavicle injuries (6 contusions, 7 acromioclavicular joint injuries (contusion, sprain/strain or dislocation), 6 dislocated shoulders, 7 clavicle fractures, 2 glenoid fractures, 1 laceration and 1 other sprain);

- Head injuries (24 concussions, 2 intracranial bleedings, 1 laceration and 1 temporal bone fracture);

- Back/neck/throat injuries (19 contusions, 3 vertebral fractures and 1 thyroid cartilage fracture);

- Hand and finger injuries (2 lacerations, 2 contusions, 2 sprains, and 12 finger fractures);

- Pelvis/hips/thigh injuries (6 contusions, 2 sprains, 1 laceration, 1 trochanter minor fracture and 5 major haematomas with 2 cases of compartment syndrome);

| | |
|---|---|
| Foot/Toes | 0.4% |
| Arm | 2.2% |
| Ankle | 2.6% |
| Chest/Flank | 3.5% |
| Wrist | 4.3% |
| Knee | 6.1% |
| Pelvis/Hips/Thigh | 6.5% |
| Hand/Fingers | 7.8% |
| Back/Neck/Throat | 10.0% |
| Head | 12.2% |
| Shoulder/Clavicle | 13.0% |
| Face | 31.3% |

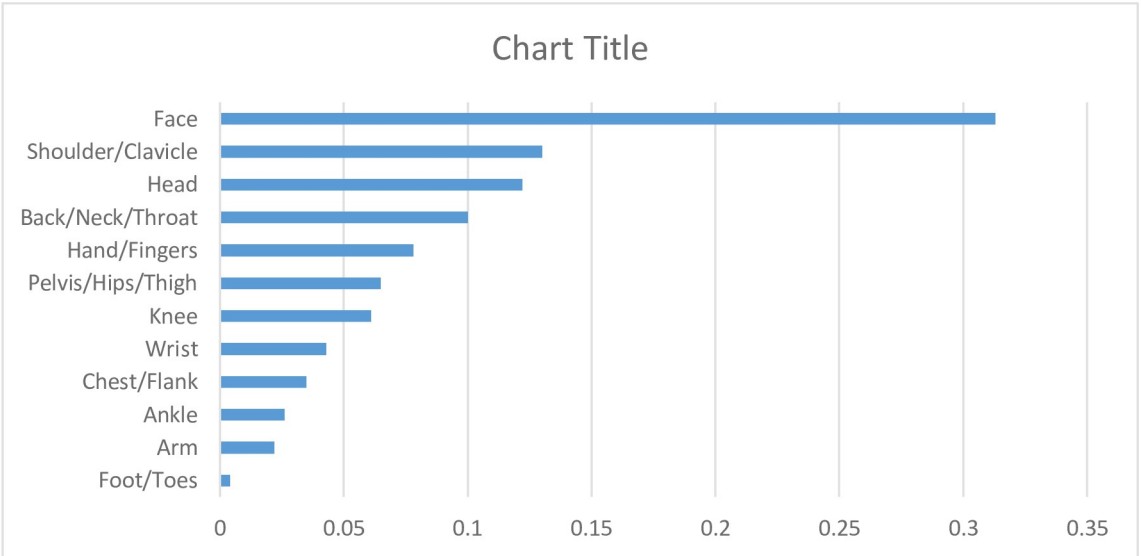

**Fig 3. Frequency of anatomical location of injury in percent.**

- Knee injuries (1 laceration, 5 contusions, 1 Bucket-handle meniscal tear, 3 lesions of the medial collateral ligament and 4 other simple sprains/strains);

- Wrist injuries (1 laceration, 2 contusions, 5 distal radius fractures, 1 trans-scaphoid perilunate fracture dislocation, 1 distal forearm fracture);

- Chest and flank injuries (7 contusions, 1 major haematoma with retroperitoneal haemorrhage from the upper pole of the kidney);

- Ankle injuries (2 contusions, 1 undefined medial malleolar fracture, 1 ankle fracture dislocation and 2 type Weber B fractures);

- Arm injuries (1 laceration, 1 humerus fracture, 1 forearm fracture and 2 sprains including a biceps tendon rupture),

- Foot/toes injury (1 contusion).

For the sake of clarity, the anatomical location of the injury can be categorised into 4 main groups, i.e. face and head; upper extremities (hand, wrist, arm, shoulder/clavicle); lower

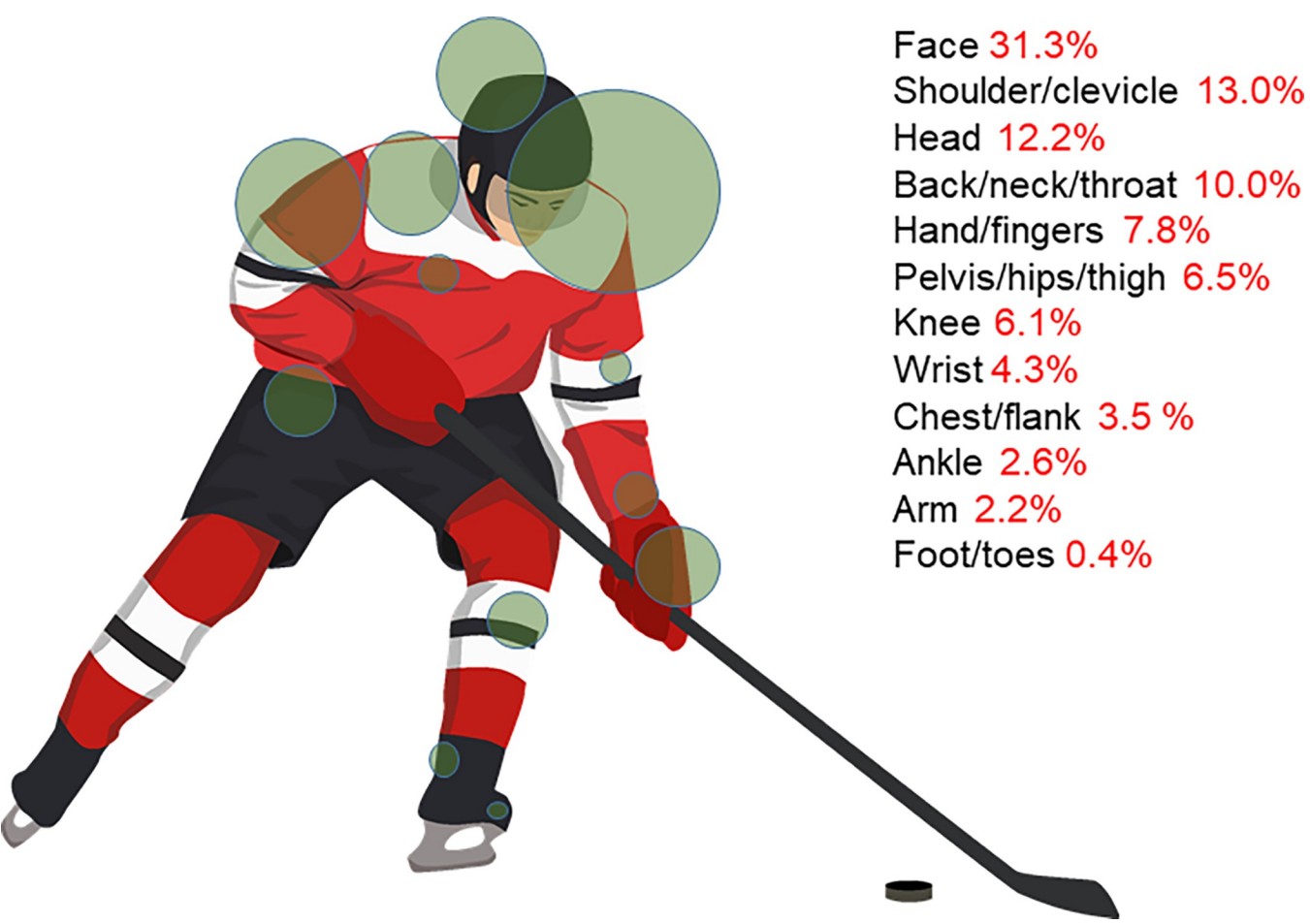

Face 31.3%
Shoulder/clevicle 13.0%
Head 12.2%
Back/neck/throat 10.0%
Hand/fingers 7.8%
Pelvis/hips/thigh 6.5%
Knee 6.1%
Wrist 4.3%
Chest/flank 3.5 %
Ankle 2.6%
Arm 2.2%
Foot/toes 0.4%

**Fig 4. Frequency of anatomical location of injury.** (Picture from www.creazilla.com).

extremities (feet/toes, ankle, knee and pelvis/hips/thigh); chest and back. The proportions are as follows: head and face (43.5%, n = 100); upper limbs (27.4%, n = 63); lower limbs (15.7%, n = 36); chest and back (13.5%, n = 31).

If the facial area was involved, the face itself was affected in 69.4% of cases (skin and bone structure) (n = 50), in 16.7% the teeth (n = 12), in 9.7% the eyes (n = 7) and in 4.2% the ears (n = 3) (Fig 5).

Within the face, the most common injuries were fractures (36.1%, n = 26) (11 mandibular fractures, 4 maxillary fractures, 4 nasal fractures, 1 frontal sinus fracture, 2 orbital blowout fractures, 4 complex maxillofacial fractures) and lacerations (29.2%, n = 21). Less frequent injuries include dental injuries (13.9%, n = 10), eye injuries (11.1%, n = 8), contusions (6.9%, n = 5) and ear/tympanum lesions (2.8%, n = 2).

As regards the *mechanism of injury*, most lesions were caused by player-player contact (37.4%, n = 86), followed by contact with the puck (24.3%, n = 56) and falls (10.9%, n = 25). In the lower areas, injuries caused by contact with the stick made up 9.1% of injuries (n = 21), injuries with the boards 5.7% (n = 13) and with the skate 2.6% (n = 6). The narratives did not contain enough information to discern the mode of injury in 10% of cases (n = 23) (Fig 6).

Within the specific case of player-player contact, more than a quarter of accidents were boarding (25.6%, n = 22).

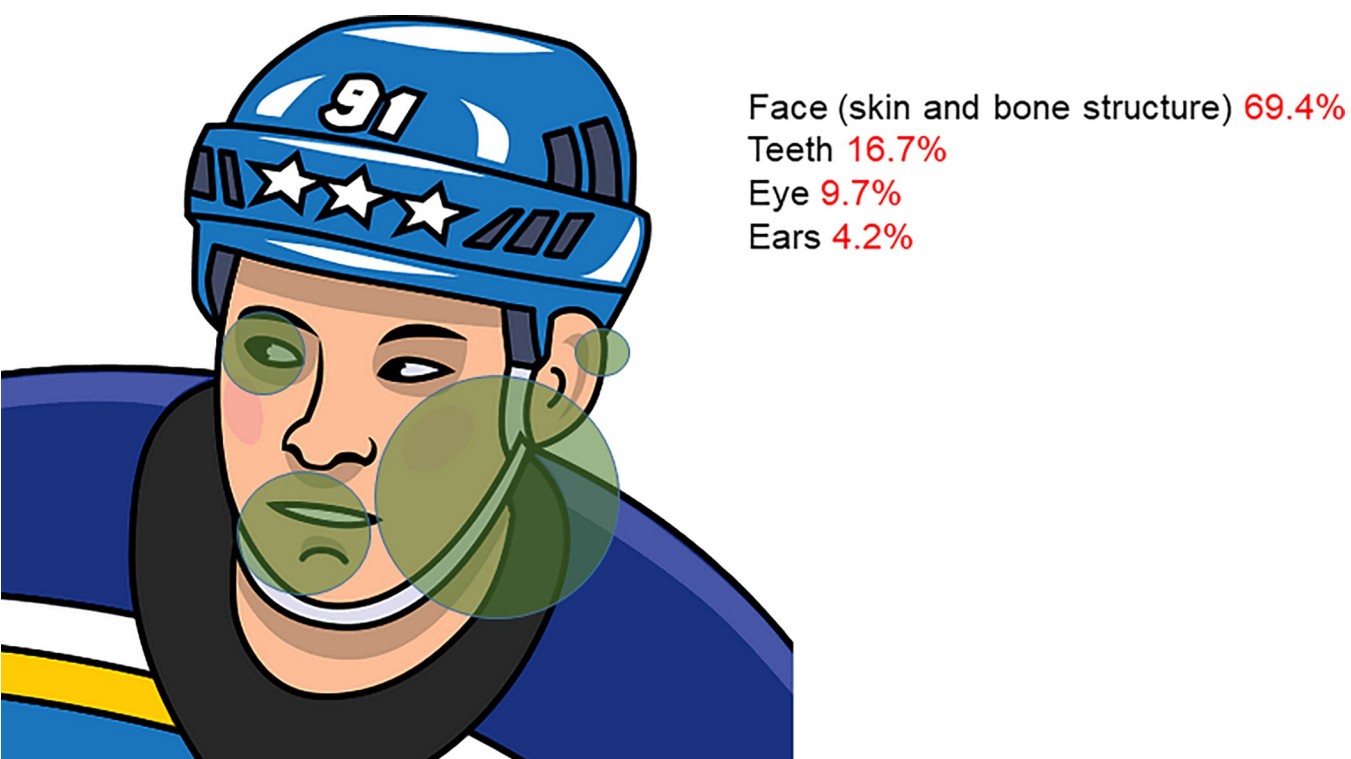

Face (skin and bone structure) 69.4%
Teeth 16.7%
Eye 9.7%
Ears 4.2%

**Fig 5. Frequency of anatomical location of injury if face involved.** (Picture from www.creazilla.com).

Falls were the mode mainly responsible for wrist injuries and intracranial bleeding. Player-player contact was mainly responsible for injuries in different body regions: head, arm, shoulder/clavicle, chest/flank, back/neck/throat, pelvis/hips/thigh, ankle; as well as for the following diagnoses: lacerations, contusion/abrasion, dislocation, concussion and major haematoma. Contact with the stick was not the main cause for any body site, but was the main cause of eye injuries. Puck contact was in turn the dominant cause of injuries for the following body regions: face, hand/fingers, foot/toes; and for the following diagnoses: laceration (together with player-player contact), fractures, intracranial bleeding (together with falls), teeth and ear/tympanum injuries. Contact with boards and skate was not the main cause of any type of injury (Table 2).

### Age-related characteristics

The anatomical location of injury, the main diagnoses and the leading mechanism of injury differ by age (Table 3).

For all age groups, the most affected part of the body was the face (34.9%, n = 44 for the 16–25-year-old cohort; 27.4%, n = 17 for the 26–35-year cohort; 26.9%, n = 7 for 36–45-year-old cohort and 25.0%, n = 4 for the ≥46-year-old cohort).

For the 16–25 year old group the most frequent diagnosis was fracture (30.2%, n = 38) and the most frequent injury mode was contact between players (42.9%, n = 54). For the 26–35 year old cohort, the most common diagnosis was contusion/abrasion (29.0%, n = 18) and the most frequent injury mode also was player-player contact (38.7%, n = 24). For the oldest cohorts, the 36–45 year old and the ≥46 year old cohorts, the most common diagnosis was fracture (26.9%, n = 7 and 31.3%, n = 5, respectively), while the most common mode was falls (30.8%, n = 8; 25%, n = 4, respectively).

| | |
|---|---|
| Falls | 10.9% |
| Player-player contact | 37.4% |
| Contact with boards | 5.7% |
| Contact with stick | 9.1% |
| Contact with puck | 24.3% |
| Contact with skate | 2.6% |
| Other/Unknown | 10.0% |

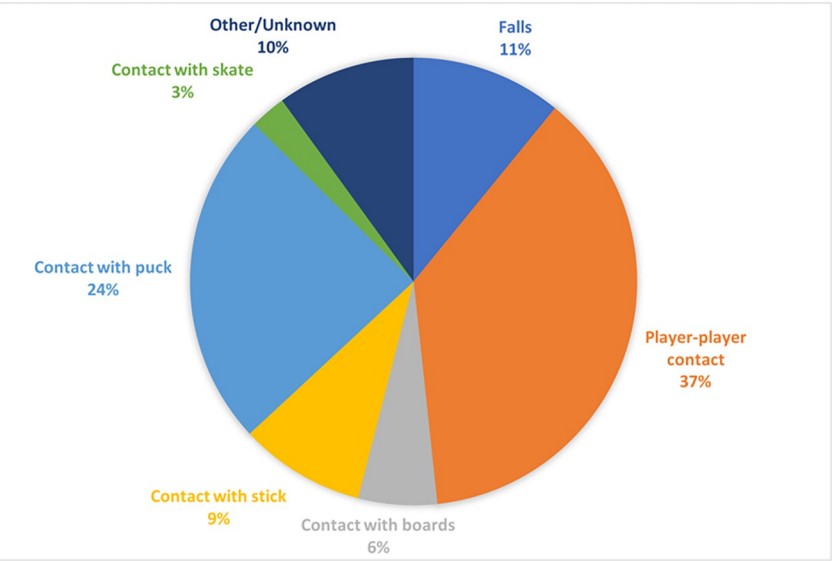

**Fig 6. Diagram showing distribution of leading mechanism of injury in percent.**

As mentioned in the methods, the statistical analysis to compare the different age groups was performed between the 16–25 year old cohort, 26–35 year old cohort and $\geq$ 36-year old cohort. There is no significant difference between the various age cohorts for the lesion location and for the diagnosis (p > 0.05). There is instead strong statistical evidence for a difference in the frequency of falls. The group of patients >36 years of age more often suffered injuries caused by falls (28.6%, n = 12), compared to the group of 26–35 years (9.7%, n = 6) and 16–25 years (5.6%, n = 7) (p-value < 0.001). Furthermore, in the cohort of patients aged >36 years, the rate of injuries from contact between players was lower (19.0%, n = 8), than in the groups of 26–35 years (38.7%, n = 24) and of 16–25 years (42.9%, n = 54) (p = 0.01813). Boarding accidents were also rarer in the oldest category: 2.4% in the >36-year old cohort (n = 1), 14.5% in the 26-35-year-cohort (n = 9) and 9.5% in the 16-25-year cohort (n = 12), but without significant difference (p = 0.1156).

## Characteristics related to ice hockey playing level

The anatomical location of injury, the frequency of diagnoses and the leading mechanism of injury differed by playing level (Table 4).

For both professional and non-professional players, the most commonly injured part of the body was the face (36.6%, n = 15; 30.2%, n = 57, respectively) and the most recurrent mechanism of injury was contact with a player (46.3%, n = 19; 35.4%, n = 67, respectively). For the professional players, the most frequent diagnosis were fracture and contusion/abrasion (both 24.4%, n = 10), for amateur players fracture (29.1%, n = 55). Neither the location of the lesion nor the reported diagnosis differed significantly between the two cohorts, (p > 0.05). Among non-professional players, there was a significantly higher prevalence of injuries caused by stick

**Table 2. Leading mechanism of injury by body part and diagnosis.** Sprain and strain include ligamentous and meniscal injuries.

| Leading mechanism of injury / Characteristics | Falls | Player-player contact | Contact with boards | Contact with stick | Contact with puck | Contact with skate | Other/ Unknown | Total |
|---|---|---|---|---|---|---|---|---|
| | n (%) | n (%) | n (%) | n (%) | n (%) | n (%) | n (%) | n (%) |
| *Body site.* | | | | | | | | |
| Face | 2 (2.8) | 12 (16.7) | 0 (0.0) | 15 (20.8) | 38 (52.8) | 3 (4.2) | 2 (2.8) | 72 (100.0) |
| Head | 6 (21.4) | 18 (64.3) | 2 (7.1) | 0 (0.0) | 2 (7.1) | 0 (0.0) | 0 (0.0) | 28 (100.0) |
| Hand/Fingers | 0 (0.0) | 5 (27.8) | 1 (5.6) | 1 (5.6) | 7 (38.9) | 1 (5.6) | 3 (16.7) | 18 (100.0) |
| Wrist | 4 (40.0) | 3 (30.0) | 0 (0.0) | 2 (20.0) | 0 (0.0) | 1 (10.0) | 0 (0.0) | 10 (100.0) |
| Arm | 1 (20.0) | 2 (40.0) | 0 (0.0) | 0 (0.0) | 0 (0.0) | 0 (0.0) | 2 (40.0) | 5 (100.0) |
| Shoulder/Clavicle | 5 (16.7) | 18 (60.0) | 4 (13.3) | 0 (0.0) | 0 (0.0) | 0 (0.0) | 3 (10.0) | 30 (100.0) |
| Chest/Flank | 2 (25.0) | 3 (37.5) | 2 (25.0) | 0 (0.0) | 1 (12.5) | 0 (0.0) | 0 (0.0) | 8 (100.0) |
| Back/Neck/Throat | 1 (4.3) | 11 (47.8) | 1 (4.3) | 3 (13.0) | 5 (21.7) | 0 (0.0) | 2 (8.7) | 23 (100.0) |
| Pelvis/Hips/Thigh | 2 (13.3) | 8 (53.3) | 0 (0.0) | 0 (0.0) | 0 (0.0) | 1 (6.7) | 4 (26.7) | 15 (100.0) |
| Knee | 2 (14.3) | 4 (28.6) | 2 (14.3) | 0 (0.0) | 1 (7.1) | 0 (0.0) | 5 (35.7) | 14 (100.0) |
| Ankle | 0 (0.0) | 2 (33.3) | 1 (16.7) | 0 (0.0) | 1 (16.7) | 0 (0.0) | 2 (33.3) | 6 (100.0) |
| Foot/Toes | 0 (0.0) | 0 (0.0) | 0 (0.0) | 0 (0.0) | 1 (100.0) | 0 (0.0) | 0 (0.0) | 1 (100.0) |
| *Main diagnosis* | | | | | | | | |
| Laceration | 3 (10.3) | 8 (27.6) | 0 (0.0) | 6 (20.7) | 8 (27.6) | 4 (13.8) | 0 (0.0) | 29 (100.0) |
| Fracture | 4 (6.2) | 23 (35.4) | 3 (4.6) | 3 (4.6) | 26 (40.0) | 1 (1.5) | 5 (7.7) | 65 (100.0) |
| Contusion/Abrasion | 9 (16.4) | 23 (41.8) | 5 (9.1) | 4 (7.3) | 8 (14.5) | 1 (1.8) | 5 (9.1) | 55 (100.0) |
| Dislocation | 2 (16.7) | 6 (50.0) | 3 (25.0) | 0 (0.0) | 0 (0.0) | 0 (0.0) | 1 (8.3) | 12 (100.0) |
| Sprain/Strain | 1 (5.9) | 4 (23.5) | 0 (0.0) | 0 (0.0) | 2 (11.8) | 0 (0.0) | 10 (58.8) | 17 (100.0) |
| Concussion | 4 (16.7) | 18 (75.0) | 2 (8.3) | 0 (0.0) | 0 (0.0) | 0 (0.0) | 0 (0.0) | 24 (100.0) |
| Intracranial bleeding | 1 (50.0) | 0 (0.0) | 0 (0.0) | 0 (0.0) | 1 (50.0) | 0 (0.0) | 0 (0.0) | 2 (100.0) |
| Dental injury | 0 (0.0) | 1 (10.0) | 0 (0.0) | 2 (20.0) | 6 (60.0) | 0 (0.0) | 1 (10.0) | 10 (100.0) |
| Eye injury | 0 (0.0) | 0 (0.0) | 0 (0.0) | 6 (75.0) | 2 (25.0) | 0 (0.0) | 0 (0.0) | 8 (100.0) |
| Ear/Tympanum lesion | 0 (0.0) | 0 (0.0) | 0 (0.0) | 0 (0.0) | 2 (100.0) | 0 (0.0) | 0 (0.0) | 2 (100.0) |
| Major Hematoma | 1 (16.7) | 3 (50.0) | 0 (0.0) | 0 (0.0) | 1 (16.7) | 0 (0.0) | 1 (16.7) | 6 (100.0) |

contact (0.0% among professional players, n = 0; 11.1% among non-professional players, n = 21) (OR 0, 95% CI (0.00–0.83), p = 0.02).

Furthermore, it is interesting to see that patients who played professionally were significantly more often submitted to surgical treatment, 29.3% (n = 12) versus 14.9% (n = 48), respectively (OR 2.35, 95% CI 0.98–5.46, p = 0.04).

**Table 3. Age-related anatomical location of injury, diagnoses and leading mechanism of injury.**

| Characteristics / Age cohort | 16–25 years | 26–35 years | 36–45 years | ≥ 46 years |
|---|---|---|---|---|
| | n (%) | n (%) | n (%) | n (%) |
| *Body site* | | | | |
| Face | 44 (34.9) | 17 (27.4) | 7 (26.9) | 4 (25.0) |
| Head | 15 (11.9) | 4 (6.5) | 6 (23.1) | 3 (18.8) |
| Hand/Fingers | 10 (7.9) | 5 (8.1) | 2 (7.7) | 1 (6.3) |
| Wrist | 5 (4.0) | 3 (4.8) | 1 (3.8) | 1 (6.3) |
| Arm | 2 (1.6) | 2 (3.2) | 0 (0.0) | 1 (6.3) |
| Shoulder/Clavicle | 18 (14.3) | 9 (14.5) | 2 (7.7) | 1 (6.3) |
| Chest/Flank | 4 (3.2) | 3 (4.8) | 0 (0.0) | 1 (6.3) |
| Back/Neck/Throat | 12 (9.5) | 8 (12.9) | 2 (7.7) | 1 (6.3) |
| Pelvis/Hips/Thigh | 8 (6.3) | 4 (6.5) | 2 (7.7) | 1 (6.3) |
| Knee | 6 (4.8) | 5 (8.1) | 2 (7.7) | 1 (6.3) |
| Ankle | 2 (1.6) | 1 (1.6) | 2 (7.7) | 1 (6.3) |
| Foot/Toes | 0 (0.0) | 1 (1.6) | 0 (0.0) | 0 (0.0) |
| **Total** | 126 (100.0) | 62 (100.0) | 26 (100.0) | 16 (100.0) |
| *Main diagnosis* | | | | |
| Laceration | 13 (10.3) | 9 (14.5) | 6 (23.1) | 1 (6.3) |
| Fracture | 38 (30.2) | 15 (24.2) | 7 (26.9) | 5 (31.3) |
| Contusion/Abrasion | 32 (25.4) | 18 (29.0) | 4 (15.4) | 1 (6.3) |
| Dislocation | 8 (6.3) | 4 (6.5) | 0 (0.0) | 0 (0.0) |
| Sprain/Strain | 7 (5.6) | 5 (8.1) | 2 (7.7) | 3 (18.8) |
| Concussion | 13 (10.3) | 4 (6.5) | 4 (15.4) | 3 (18.8) |
| Intracranial bleeding | 1 (1.0) | 0 (0.0) | 1 (4.0) | 0 (0.0) |
| Dental injury | 5 (4.0) | 2 (3.2) | 1 (3.8) | 2 (12.5) |
| Eye injury | 6 (4.8) | 2 (3.2) | 0 (0.0) | 0 (0.0) |
| Ear/Tympanum lesion | 1 (0.8) | 1 (1.6) | 0 (0.0) | 0 (0.0) |
| Major Hematoma | 2 (1.6) | 2 (3.2) | 1 (3.8) | 1 (6.3) |
| **Total** | 126 (100.0) | 62 (100.0) | 26 (100.0) | 16 (100.0) |
| *Leading mechanism of injury* | | | | |
| Falls | 7 (5.6) | 6 (9.7) | 8 (30.8) | 4 (25.0) |
| Player-player contact | 54 (42.9) | 24 (38.7) | 6 (23.1) | 2 (12.5) |
| Contact with boards | 7 (5.6) | 4 (6.5) | 1 (3.8) | 1 (6.3) |
| Contact with stick | 12 (9.5) | 6 (9.7) | 3 (11.5) | 0 (0.0) |
| Contact with puck | 30 (23.8) | 16 (25.8) | 5 (19.2) | 5 (31.3) |
| Contact with skate | 5 (4.0) | 1 (1.6) | 0 (0.0) | 0 (0.0) |
| Other/Unknown | 11 (8.7) | 5 (8.1) | 3 (11.5) | 4 (25.0) |
| **Total** | 126 (100.0) | 62 (100.0) | 26 (100.0) | 16 (100.0) |

## Patients presenting several times to the ER

13 patients, corresponding to 5.7% of the included cases, presented to the emergency room several times. Of these, 12 patients attended the hospital twice and one patient three times. For the vast majority, i.e. 92.3%, the diagnosis was different from that reported the previous times (n = 12). One patient had the same diagnosis twice, i.e., an acromioclavicular joint injury. Of the patients presenting several times, 46.2% were professional players and the remaining 53.8% were non-professional players (respectively, n = 6 and n = 7). Thus, the percentage of

**Table 4. Anatomical location of injury, diagnoses, leading mechanism of injury and treatment related to ice hockey playing level (professional versus non-professional).**

| Characteristics | Professional players | Non-professional players |
|---|---|---|
| | n (%) | n (%) |
| *Body site* | | |
| Face | 15 (36.6) | 57 (30.2) |
| Head | 6 (14.6) | 22 (11.6) |
| Hand/Fingers | 3 (7.3) | 15 (7.9) |
| Wrist | 0 (0.0) | 10 (5.3) |
| Arm | 0 (0.0) | 5 (2.6) |
| Shoulder/Clavicle | 4 (9.8) | 26 (13.8) |
| Chest/Flank | 1 (2.4) | 7 (3.7) |
| Back/Neck/Throat | 4 (9.8) | 19 (10.1) |
| Pelvis/Hips/Thigh | 5 (12.2) | 10 (5.3) |
| Knee | 3 (7.3) | 11 (5.8) |
| Ankle | 0 (0.0) | 6 (3.2) |
| Foot/Toes | 0 (0.0) | 1 (0.5) |
| **Total** | 41 (100.0) | 189 (100.0) |
| *Main diagnosis* | | |
| Laceration | 3 (7.3) | 26 (13.8) |
| Fracture | 10 (24.4) | 55 (29.1) |
| Contusion/Abrasion | 10 (24.4) | 45 (23.8) |
| Dislocation | 3 (7.3) | 9 (4.8) |
| Sprain/Strain | 4 (9.8) | 13 (6.9) |
| Concussion | 5 (12.2) | 19 (10.1) |
| Intracranial bleeding | 1 (2.4) | 1 (0.5) |
| Dental injury | 2 (4.9) | 8 (4.2) |
| Eye injury | 0 (0.0) | 8 (4.2) |
| Ear/Tympanum lesion | 1 (2.4) | 1 (0.5) |
| Major Haematoma | 2 (4.9) | 4 (2.1) |
| **Total** | 41 (100.0) | 189 (100.0) |
| *Leading mechanism of injury* | | |
| Falls | 2 (4.9) | 23 (12.2) |
| Player-player contact | 19 (46.3) | 67 (35.4) |
| Contact with boards | 1 (2.4) | 12 (6.3) |
| Contact with stick | 0 (0.0) | 21 (11.1) |
| Contact with puck | 11 (26.8) | 45 (23.8) |
| Contact with skate | 1 (2.4) | 5 (2.6) |
| Other/Unknown | 7 (17.1) | 16 (8.5) |
| **Total** | 41 (100.0) | 189 (100.0) |
| *Treatment* | | |
| Surgical | 12 (29.3) | 28 (14.8) |
| Conservative | 29 (70.7) | 160 (84.7) |
| Unknown | 0 (0.0) | 1 (0.5) |
| **Total** | 41 (100.0) | 189 (100.0) |

professionals was greater here than the overall value of 17.8%. The small number of patients with multiple consultations in the emergency department did not allow us to draw any statistically relevant consideration.

## Concussion

Concussion was reported in 10.4% of patients (n = 24). 95.8% of them were treated on an outpatient basis (n = 23), whereas one patient, corresponding to 4.2%, was hospitalised for analgesia and monitoring. 75.0% were caused by contact with another player (n = 18), 16.7% by falls (n = 4) and 8.3% by contact with the boards (n = 2). In the specific case of a player-player contact, 22.2% (n = 4) were boarding. There is no significant difference between the various age cohorts for concussion (p > 0.05). Concussion was present in 12.2% of patients playing professionally (n = 5) and in 10.1% of those playing non-professionally (n = 19), but without a significant difference (OR 1.24, 95% CI (0.34–3.74), p = 0.78).

## Discussion and limitations

This study has made it possible to take a look at the nature of ice hockey accidents in a Level-1 Emergency Centre in Switzerland over a 7-year period. In our study, the type of lesion was clearly defined through a medical consultation in the emergency room, so the accuracy of diagnoses is assumed. The health care data collected are reliable data, and do not incorporate the possible recall bias that can result from self-reporting.

97% of the patients were men and 3% women. The fractions of male and female patients were similar to that in other analyses [24, 25].

Our findings are consistent with other studies, in which most of the injuries were minor and admission to the intermediate care unit or intensive care unit was only necessary in a few cases [7, 13, 26, 27]. Most of our patients received treatment as outpatients, and came to 79.1%. In previous studies involving United States emergency departments, Morissey et al. and Deits et al. observed outpatient treatment rates higher than 99% [24] and 98% [25], respectively. The different rate of hospital admissions that we found could be explained by the fact that our clinic is also responsible for the surrounding regions for the treatment of more complex lesions that require specialised disciplines, not present elsewhere (e.g., craniomaxillofacial surgery, ophthalmology, otolaryngology) and which more commonly require hospitalisation. Patients requiring this specialist care are often transferred to us from other clinics and other physicians.

The most common diagnoses we observed were fractures, followed by contusion/abrasion, laceration and concussion. Caputo et al. also found that fractures were the most common injury in a prospective observational cohort study among adult recreational ice hockey players [28]. Other studies reported lacerations [12, 24, 25, 29], contusions [9, 13], sprains/strains [15, 26] or concussions [3] as the most frequent type of injury. Those differences are likely to be due to non-uniform patterns of defining injury, descriptions of body locations, as well as different age, playing level and study designs, so that direct comparison is difficult [8, 16].

It is important to note that for the analysis of the final data we took into account only the main diagnoses, i.e., those with the most clinical relevance. This inevitably led to an underestimation of minor diagnoses such as lacerations and contusions. For example, in the case of an open fracture or of concussion concomitant to a head laceration, the diagnosis of laceration would not be considered. In addition, as Bern University Hospital is a specialised centre of reference for the region, as mentioned above, it is possible that more common diagnoses, like lacerations or concussions, may be underrepresented as they are also treatable in other medical centres, with over-representation of more specialist diagnoses, like craniomaxillofacial fractures or eye injuries. The Inselspital is the only referral centre in the region for severe craniofacial trauma. However, the incidence of sports-related facial bone fractures is significant, and along with cycling, skiing, and soccer, ice hockey is one of the sports that causes the most maxillofacial injuries [30].

A 9-year prospective study at the International Ice Hockey Federation (IIHF) Under-20 World Championships analysed injuries in junior hockey. Hockey injuries to the head and face made up 39%, to the upper body 29%, to the lower body 24% and to the spine or trunk 9% [12], with proportions not far from those found by us.

In our observation, injuries were most commonly in the face, the shoulder/clavicle and the head. The most frequent injuries involving the face were fractures and lacerations, followed by dental injuries. The face was primarily affected regardless of age group, as shown by other studies carried out on different categories of players [5, 12, 24–26, 28, 29]. Some studies considered the face and head as a single region, and reported that this was the most affected region [6, 12, 17]. The shoulder and clavicle were the most frequently injured parts of the upper extremities, in accordance with other analyses [12, 29].

Because it is so often affected, special attention must be paid to the facial region and prevention measures should concentrate on face protection equipment. The positive protective effects of a mouth guard and a face mask are well known [31]. In the cases we analysed, it was seldom reported whether the patient was wearing protective equipment or not and equally important, whether it was worn correctly, which prevented us from drawing conclusions about it. However, as most of the patients were recreational players, it is likely that face protection was not always sufficient. The rate of use of mouth guards by players has been reported to be between 13.3% and 67.3%, and these guards tend to be omitted during practice [32, 33]. An observational prospective cohort study of injuries among adult recreational ice hockey players found that all facial injuries reported were in players who were not wearing facial protection [28]. In another study, 20% of amateur hockey players reported that they had not been wearing any facial protection at the time of injury. The same study reported a higher probability of incurring a facial laceration or facial bone fracture in players who did not use face protection [34]. In an emergency department in Canada, players wearing helmets without facemasks were more frequently injured by contact with sticks or pucks to the head, face, and neck [35]. The use of full-face visors has also been shown to be associated with a significantly reduced risk of sustaining facial and dental injuries without increasing the risk of neck injury, concussion or other injuries [27, 36] and in a cohort study MJ Stuart et al. observed that players who did not wear any protection were injured at more than twice the rate of those who wore partial protection and almost seven times more than those who wore full protection [36].

Despite the efforts that have been made since the 1990s to develop injury prevention in ice hockey, a wide variety of injuries, especially to the face, continue to be encountered in emergency rooms. This shows that there is still work to be done in this direction. Our findings indicate that the protection of the face is inadequate. An improvement in this regard could lead to a decrease in injuries occurring on the ice hockey rink. Proper face protection should be recommended and implemented by sports federations and all clinicians who interact with athletes. The same advice applies to the requirement of using a mouth guard in all playing circumstances and categories, not only in the under-20 categories as currently regulated [18]. Moreover, in our study most incidents occur at a non-professional level, where there is less strict and uniform regulation of the game rules. Therefore, prevention at this level through increased information is all the more important, and probably still insufficient.

We found that the most frequent mechanisms leading to injuries were player-to-player contact, followed by puck contact and falls. In the specific case of player-player contact, more than a quarter consisted of body checking against the boards. Boarding was also relevant to concussions, and caused 22.2% of these cases. Other studies at different levels described contact between players as a primary source of injury, [3, 10, 17, 26, 28, 37], especially through body checking [12, 13, 29, 38, 39].

In agreement with similar observations, the main age group affected by injuries was the youngest, here aged between 16–25 years old, with decreasing incidence of injuries as age increases [24, 25]. We observed a similar decreasing trend with age for injuries caused by player-to-player contact. The lower frequency of injury through contact between players in the older age groups is probably due to the fact that there are fewer active players with increasing age, and that they play at a less competitive level and skate at slower velocities. Younger athletes may be more energetic and more competitive. The 36+-year-old cohort had a significantly higher proportion of injuries caused by falls. The same was found by Morrissey et al. [24]. This is probably due to decreasing practice in ice hockey, as well as diminished reflexes with increasing age.

Body checking is related to an increased risk of injury at different age levels [6, 15, 16, 40–42], with a significantly increased risk of all game-related injuries and concussions in youth [5, 14]. In addition, injury by a body check would be a risk factor for increased time loss caused by injury [39]. As this plays a central role in injury mechanisms, it makes sense to ask whether more restrictions should be imposed on body checking. In our study, the narratives did not allow differentiation between collisions and body checking. While it is plausible that limiting body checking could decrease injuries, but this remains unproven.

In our analysis most of the patients were non-professional. We founded no substantial differences between professional and amateur players in the type of diagnosis and location of injuries. Among amateur players there was a significantly higher prevalence of injuries caused by stick contact. At the professional level, protective equipment is worn more consistently, which may explain the better protection from a stick injury. However, according to this hypothesis, a difference should also have been shown for puck injuries. Another explanation could be that more experienced players have a greater command of the stick and the rules of the game or maybe players who play in professional leagues are more protected from contact as their referees are better trained, but these are only speculations. We also noted that patients who played professionally were significantly more likely to undergo surgical treatment. We did not find any other studies that made this comparison. Surgery may more often be considered in professional players, so that they can more rapidly recover their pre-injury level of sports activity. It can be assumed that both the physical conditions, due to the different intensity of training, and the protective equipment are significantly different between professional and amateur ice hockey players, which makes the conclusions susceptible to bias. Future studies should report in detail on the protective material worn in order to draw appropriate conclusions.

Concussions continue to be a serious problem in the ice hockey population, and in our study was one of the four most common injuries, with 10.4% of cases, a value that is not huge but still clinically relevant. A previous prospective, observational study carried out in the International Ice Hockey Federation World Junior under-20 and under-18 Championship reported similar percentage findings [12].

In our case, the large majority of concussions were caused by contact between players (75.0%) and approximately one fifth were due to boarding. Further studies reported contact with another player as the main cause of concussion [10, 12]. In intercollegiate ice hockey players, the use of a full-face shield compared with half face shield, as well as wearing a mouth guard significantly reduced the severity of concussion, as measured by time lost from competition [43]. A mouth guard not only protects the teeth, but also stabilises the jaw and can reduce the severity of a craniocerebral trauma [44]. Finally, concussion rates are more consistent in minor leagues that allow body checking [6, 40, 42].

This potentially life-threatening injury has high rates of morbidity. Multiple concussions are linked to a risk of developing migraine headaches, mental health issues, such as depression

and chronic traumatic encephalopathy [23]. The ban on hitting the head does not seem to be a sufficient measure, which indicates that further changes are needed [41]. A program called "Respect My Head" was introduced in 2011 by the Swiss Hockey League. Through videos and explanatory leaflets, it aimed to raise awareness about concussion and its possible prevention. Further informative programs should also be planned for the amateur population as they are important preventive measures.

The numerous diagnostic investigations, hospital stays and operations, as listed in our study, as well as the loss of earnings resulting from an injury, also have economic consequences. A previous investigation in the National Hockey League (Canada) showed a particularly large economic impact for head, neck, shoulder, foot and leg injuries. The estimated salary loss was US$218 million per year due to injury-associated lost time, US$42.8 million for concussions alone [45].

There are limitations to this study in addition to those already mentioned. As it is a retrospective study, there is inevitably a lack of data. An example of this is that in 10% of cases, the mechanism of injury was unknown or could not be categorised. The number of patients analysed allowed some interesting overall considerations. However, when patients were compared between various subgroups, the sample of patients of the various subgroups became small, limiting the possibilities for statistical analysis. In order to minimise this problem, a test has been used that is also valid for small samples.

The analysed cases only refer to consultations in our emergency department. Patients presenting to other hospitals and clinics, primary care physicians, team physicians and specialists do not appear here. The fact that professional teams have their own team doctor, who can deal independently with minor traumas like lacerations, also contributes to a reduction in the number of patients considered. Data from patients who have not seen a physician are also missing. Despite these limitations, the documentation of ER consultations allowed us to collect several data and draw some interesting considerations. This study helps us to understand and characterise the injuries occurring in ice hockey in our latitudes.

Prospective studies should be performed to improve the characterisation of ice hockey injuries and to capture their long-term consequences. A deeper understanding of accidents and injury types must be the basis for appropriate patient care and evidence-based prevention strategies. The epidemiology of sports injuries has led and will continue to lead to rule changes, improved equipment and targeted training. Emphasis should be placed on the role of player-player contact and the compliance of facial protection structures.

A certain degree of aggression is part of ice hockey and it is not possible to avoid any risk within the sport. But the goal is to put the acquired evidence into practice to avoid unnecessary risks and safeguard the health of athletes. In addition, evidence-based information should increase the sensitivity of players and the general public, reducing their resistance to further changes in rules and equipment.

## Conclusion

Contact between players and a facial involvement appear to be central in ice hockey injuries. Prevention initiatives should focus on strategies that limit player-to-player contact. As authors of this study, we strongly recommend increased facial protection.

## Author Contributions

**Conceptualization:** Aristomenis Exadaktylos, Jolanta Klukowska-Rötzler.

**Data curation:** Jolanta Klukowska-Rötzler.

**Formal analysis:** Viola Gilardi, Spyridon Kotsaris.

**Methodology:** Viola Gilardi, Spyridon Kotsaris, Jolanta Klukowska-Rötzler.

**Supervision:** Aristomenis Exadaktylos.

**Validation:** Spyridon Kotsaris, Jolanta Klukowska-Rötzler.

**Writing – original draft:** Viola Gilardi.

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
