## [Decision Letter · Decision Letter 0]

17 Jan 2022

PONE-D-21-34814Injury patterns of non-fatal accidents related to ice hockey, an analysis of 7 years of admission to a Level-1 Emergency Centre in SwitzerlandPLOS ONE

Dear Dr. Klukowska-Rötzler,

Thank you for submitting your manuscript to PLOS ONE. After careful consideration, we feel that it has merit but does not fully meet PLOS ONE’s publication criteria as it currently stands. Therefore, we invite you to submit a revised version of the manuscript that addresses the points raised during the review process. Ice hockey is the preferred team sport of the academic editor. Therefore I would be extremely happy to get your revised manuscript. Three reviewers very familiar with ice hockey injuries made useful recommendations to revise the manuscript. Please, follow their ideas. I am convinced that the impact of your study can be much improved.

We look forward to receiving your revised manuscript.

Kind regards,

Hans-Peter Simmen, M.D., Professor of Surgery

Academic Editor

PLOS ONE

Journal Requirements:

2. Please include your tables as part of your main manuscript and remove the individual files. Please note that supplementary tables (should remain/ be uploaded) as separate "supporting information" file.

3. You indicated that you had ethical approval for your study. In your Methods section, please ensure you have also stated whether you obtained consent from parents or guardians of the minors included in the study or whether the research ethics committee or IRB specifically waived the need for their consent.

4. We note that Figure 1 includes an image of a [patient / participant / in the study].

If you are unable to obtain consent from the subject of the photograph, you will need to remove the figure and any other textual identifying information or case descriptions for this individual

5. We note that you have stated that you will provide repository information for your data at acceptance. Should your manuscript be accepted for publication, we will hold it until you provide the relevant accession numbers or DOIs necessary to access your data. If you wish to make changes to your Data Availability statement, please describe these changes in your cover letter and we will update your Data Availability statement to reflect the information you provide

Reviewers' comments:

Reviewer's Responses to Questions

**Comments to the Author**

1. Is the manuscript technically sound, and do the data support the conclusions?

Reviewer #1: No

Reviewer #2: Yes

Reviewer #3: Partly

2. Has the statistical analysis been performed appropriately and rigorously? 

Reviewer #1: No

Reviewer #2: Yes

Reviewer #3: I Don't Know

3. Have the authors made all data underlying the findings in their manuscript fully available?

Reviewer #1: Yes

Reviewer #2: Yes

Reviewer #3: No

4. Is the manuscript presented in an intelligible fashion and written in standard English?

Reviewer #1: Yes

Reviewer #2: Yes

Reviewer #3: No

5. Review Comments to the Author

Reviewer #1: You have a done a good study. Your discussion is well done, and you have also found where we have to increase our investigations in order to ameliorate the safety of the ice hockey players not only in the professional but also in the amateur leagues.

However, there are some points which have to be ameliorates in order to publish such a study.

Concerning „facial region and prevention measures”, Lines 463-465

In order to analyze the relationship between an injury and the protective equipment it is not only mandatory to know if the players were wearing the equipment at all, but also if it was wearied correctly, otherwise we are making wrong conclusion.

Concerning “facial protection”, Lines 481-483

The precise analyze of the relationship between facial protection and injury is one of the most important requirements in the prevention of these injuries and it should be done very precisely! Your opinion is not completely supported by our data.

Concerning “Body checking is related to an increased risk of injury at different age levels”, Line 503-506

Here it would be better if you’d have done an analyze between the players ages differently, first of between U17/U18 and NL/SL and SWHL A, and if you’d have done an analyze between the role of body weight and large size differentials between players in some age groups.

Concerning “more restrictions should be imposed on body checking”, Lines 507-509

You are comparing four different group of players’ ages, which make the result and first of the discussion and conclusions concerning the problems of body checking very difficult.

Concerning “professional and amateur players”, Line 511-520

Your definition of professional and non-professional is not optimal to make such conclusions.

Concerning “concussion”, Lines 522-532

Here, we are missing a clear definition of concussion. In order to analyze the problems of concussions, you should use the most internationally recognized definition of concussion such as the definition of the “Sport Concussion Assessment Tool”, which has been published by the Consensus statement on concussion in sport—the 5th international conference on concussion in sport held in Berlin, October 2016, McCrory P, et al. Br J Sports Med 2017.

Reviewer #2: The study shows a nice overview of injuries in ice hockey. On the other hand, however, the data are mostly only listed in a very general and tabular way. There is a lack of linkage with each other.

Examples: Which diagnoses are correlated with which body parts (e.g., upper extremity fractures, facial lacerations, etc.). Which diagnoses are associated with which treatment (e.g., how many fractures were treated conservatively and which were treated surgically?). The question of severity and outcome arises: who had to be hospitalized with which diagnoses and for how long. The socio-economic aspect is mentioned, but only applied to the direct treatment costs; this study lacks data on hospitalization/duration, loss of work and long-term problems in relation to specific injuries and mechanisms (example: cruciate ligament injuries with months of absence versus banal laceration, see also Fig. 6 with illustration of facial injuries (skin and bone structure) 69.4%: while skin injuries are mostly uncomplicated here, facial fractures are nevertheless associated with considerably more morbidity).

This survey also does not indicate whether there are specific ice hockey injuries that are disproportionately represented in this specific sport because the data are so general. The bias between center function for professional players and general medical care for amateur players is discussed, but is not evident in the overview because no specific information on diagnoses and treatments is available.

Reviewer #3: The present study focuses on non-fatal injury patterns in ice hockey players. Within the framework of a retrospective, descriptive database evaluation, the injury patterns in ice hockey players who were treated at a level 1 trauma center over a period of 7 years are analyzed.The design of the study is highly susceptible to bias, not only due to its retrospective nature. As part of the evaluation, the injuries of professional and amateur ice hockey players are evaluated. It can be assumed that both the physical conditions, due to the different intensity of training sessions, and the protective equipment will be significantly different between these two groups. The authors marginally address this point of criticism, but a corresponding conclusion regarding the adaptation of the study design is missing. Also, the almost exclusive inclusion of male ice hockey players (male percentage 97%) should lead to an adjustment of the study design. The Introduction, and here in particular the first and third paragraphs, consists predominantly of well-known lexicon knowledge and prose and should be clearly shortened to the essentials. On page 4, lines 95-96, the authors state that in most of the studies published on the topic under investigation, the study population consisted primarily of young subjects. In the present study, however, the study population also consists of just under 55% adolescent subjects. This raises the question of what new scientific findings can be derived from the present study.

Generally:

- The sport meant on page 3 line 66 should be called "soccer" and not (American) "football".

- Also on page 3 line 79 there is a spelling mistake: it should read "definitions".

- Overall, the manuscript should be revised for spelling and English grammar.

Suggestion:

The authors should revise their analysis to focus on one study population. They should also explain what new scientific evidence the results of their study provide compared with the existing literature on sports injuries in ice hockey.

6. PLOS authors have the option to publish the peer review history of their article (what does this mean?). If published, this will include your full peer review and any attached files.

Reviewer #1: No

Reviewer #2: **Yes: **Walter Kistler

Reviewer #3: No

---

## [Author Response · Author response to Decision Letter 0]

9 May 2022

Bern, March 28, 2022

PLOS ONE

Manuscript ID PONE-D-21-34814

Dear Editor

We appreciate your interest and your attentive and helpful peer-review of this paper: 

"Injury patterns of non-fatal accidents related to ice hockey, an analysis of 7 years of admission to a Level-1 Emergency Centre in Switzerland"

We thoroughly considered your comments and these gave us the opportunity to im-prove and extend the scope of our paper. 

Thank you to the reviewers for critically looking at the study and making suggestions for improvement:

1.Please ensure that your manuscript meets PLOS ONE's style requirements, includ-ing those for file naming. The PLOS ONE style templates can be found at

2. Please include your tables as part of your main manuscript and remove the individ-ual files. Please note that supplementary tables (should remain/ be uploaded) as sepa-rate "supporting information" file.

Answer: 1-2: We have adapted the manuscript to PLOS ONE's style requirements.

3. You indicated that you had ethical approval for your study. In your Methods section, please ensure you have also stated whether you obtained consent from parents or guardians of the minors included in the study or whether the research ethics commit-tee or IRB specifically waived the need for their consent.

Answer: The General Consent (GC) was introduced at our University Emergency Centre in February 2015. During their stay at the University Emergency Centre, many patients cannot reasonably be approached regarding research or General Consent. Some of the patients are unresponsive or not in a state to give valid consent (medication influence, shock). However, since February 2016 our staff has been working to obtain a signed GC from as many patients as possible. 50% patients were approached monthly regarding GC. The percentage of refusal of GC is 5%. Half the patients are documented in our sys-tem as Granted and the rest as Informed. If there is a documented GC refusal, or contra-diction to use patient data, these are not recorded.

We consider our group of patients to be particularly vulnerable, so that fewer GCs were probably obtained here than in the average emergency patients. 

The research ethics committee waived to further individual informed consent for this study. The proposed study poses at most minimal risks to patients, but offers potential benefit to future patients.

In our case, according to the protocol, we only include 16-year-old patients in the study. Younger patients go to a separate children’s emergency ward. Therefore, parental con-sent is not required; in Switzerland, all persons aged 16 and over are fully responsible.

4. As per the PLOS ONE policy (http://journals.plos.org/plosone/s/submission-guidelines#loc-human-subjects-research) on papers that include identifying, or poten-tially identifying, information, the individual(s) or parent(s)/guardian(s) must be in-formed of the terms of the PLOS open-access (CC-BY) license and provide specific permission for publication of these details under the terms of this license. Please download the Consent Form for Publication in a PLOS Journal (http://journals.plos.org/plosone/s/file?id=8ce6/plos-consent-form-english.pdf). The signed consent form should not be submitted with the manuscript, but should be se-curely filed in the individual's case notes. Please amend the methods section and eth-ics statement of the manuscript to explicitly state that the patient/participant has provid-ed consent for publication: “The individual in this manuscript has given written in-formed consent (as outlined in PLOS consent form) to publish these case details”.

Answer: This photo was taken by an accredited professional photographer, who has permission to photograph athletes. We attach to the materials the consent of the photog-rapher to provide the photo. The athletes shown in the picture are not patients included in the research group, this is only an example of a dangerous maneuver during an ice hockey match. If the editor orders it, you can put black stripes in front of the athletes' eyes.

5. We note that you have stated that you will provide repository information for your da-ta at acceptance. Should your manuscript be accepted for publication, we will hold it until you provide the relevant accession numbers or DOIs necessary to access your data. If you wish to make changes to your Data Availability statement, please describe these changes in your cover letter and we will update your Data Availability statement to reflect the information you provide.

Answer: Due to the decision of the ethics committee, we are allowed to publish the re-sults of the analysis, but we are not allowed to give access to all patient data.

Reviewer #1:

- Comment: Concerning “Body checking is related to an increased risk of injury at different age levels”, Line 503-506. Here it would be better if you’d have done an analysis between the players ages differently, first of between U17/U18 and NL/SL and SWHL A, and if you’d have done an analysis between the role of body weight and large size differentials between players in some age groups.

o Answer: This type of analysis is certainly interesting. However, we are concerned that our data do not allow this type of categorization, due to the limited number of patients within the subgroups. For instance, in the case of SWHL A there are only two subjects. 

The weight and height of the players was only reported in rare single cases. Therefore, no conclusions could be drawn in this regard. However, we will definitely take this into account in a possible prospective future study.

- Comment: Concerning “more restrictions should be imposed on body check-ing”, Lines 507-509. You are comparing four different group of players’ ages, which make the result and first of the discussion and conclusions concerning the problems of body checking very difficult.

- 

o Answer: We are aware of our limitations in order to draw targeted conclu-sions regarding body checking. We discussed the limitations in question extensively in the dedicated section of the manuscript (lines 505-511).

We found that the most frequent mechanisms leading to injuries were player-to-player contact. Many studies have shown that body checking is associated with an increased risk of injury at different age levels. While a decrease in injuries by limiting body checking is plausible, a direct con-clusion regarding the role of body checking is not possible on the basis of our results.

- Comment: Concerning “professional and amateur players”, Line 511-520. Your definition of professional and non-professional is not optimal to make such con-clusions.

o Answer: We are conscious of the relatively arbitrary division into profes-sional and amateur players. The division was made from the level at which players usually start to receive payment. If the comparison is not considered relevant, this chapter can be removed.

Reviewer #2: The study shows a nice overview of injuries in ice hockey. On the other hand, however, the data are mostly only listed in a very general and tabular way. There is a lack of linkage with each other.

Examples: Which diagnoses are correlated with which body parts (e.g., upper extremity fractures, facial lacerations, etc.). Which diagnoses are associated with which treat-ment (e.g., how many fractures were treated conservatively and which were treated surgically?). The question of severity and outcome arises: who had to be hospitalized with which diagnoses and for how long. The socio-economic aspect is mentioned, but only applied to the direct treatment costs; this study lacks data on hospitaliza-tion/duration, loss of work and long-term problems in relation to specific injuries and mechanisms (example: cruciate ligament injuries with months of absence versus banal laceration, see also Fig. 6 with illustration of facial injuries (skin and bone structure) 69.4%: while skin injuries are mostly uncomplicated here, facial fractures are neverthe-less associated with considerably more morbidity).

This survey also does not indicate whether there are specific ice hockey injuries that are disproportionately represented in this specific sport because the data are so gen-eral. The bias between center function for professional players and general medical care for amateur players is discussed, but is not evident in the overview because no specific information on diagnoses and treatments is available.

- Answer: The diagnoses related to the body parts (lines 296-315 and 332-337), the rate of fractures treated conservatively and surgically, the surgically treated diagnoses and fractures (lines 255-262), as well as the diagnoses that required hospitalisation with the mean length of hospital stay have been added to the text (lines 224-232). 

Unfortunately, the data concerning the incapacity for work and the long-term impairments are incomplete, so it was not possible to develop a proper discus-sion about it. A targeted collection in this sense, with a focus on long-term prob-lems, including the loss of work, is being considered as a future prospective study of our University Emergency Centre.

Reviewer #3: The present study focuses on non-fatal injury patterns in ice hockey players. Within the framework of a retrospective, descriptive database evaluation, the injury patterns in ice hockey players who were treated at a level 1 trauma center over a period of 7 years are analyzed. The design of the study is highly susceptible to bias, not only due to its retrospective nature. As part of the evaluation, the injuries of profes-sional and amateur ice hockey players are evaluated. It can be assumed that both the physical conditions, due to the different intensity of training sessions, and the protec-tive equipment will be significantly different between these two groups. The authors marginally address this point of criticism, but a corresponding conclusion regarding the adaptation of the study design is missing. Also, the almost exclusive inclusion of male ice hockey players (male percentage 97%) should lead to an adjustment of the study design. The Introduction, and here in particular the first and third paragraphs, consists predominantly of well-known lexicon knowledge and prose and should be clearly shortened to the essentials. On page 4, lines 95-96, the authors state that in most of the studies published on the topic under investigation, the study population consisted primarily of young subjects. In the present study, however, the study population also consists of just under 55% adolescent subjects. This raises the question of what new scientific findings can be derived from the present study.

- Answer: The likely difference in physical preparation and in the protective mate-rial used are important and rightful points of discussion. These have been add-ed to the text (lines 572-577).

Our study reports almost exclusively male players. However, studies concerning other emergency centres have reported rates of male and female patients not far from ours [10,12]. Increasing the observation over several years or several multi-ple centres would lead to a substantial increase in the number of female pa-tients. On the contrary, as males represent the vast majority of (Swiss) hockey players, future analyses could be made exclusively on male patients.

The introduction has been shortened. 

Although our study also deals with predominantly young patients, there is rela-tively limited literature on the population of emergency centres, particularly in our country.

- Answer to the other comments: a change was made directly in the text of the article.

We believe that after the current revisions our paper is ready to be accepted and pub-lished in PLOS ONE.

We wish you best regards and good health.

Viola Gilardi

Jolanta Klukowska-Rötzler

---

## [Editor Report · Decision Letter 1]

11 May 2022

Injury patterns of non-fatal accidents related to ice hockey, an analysis of 7 years of admission to a Level-1 Emergency Centre in Switzerland

PONE-D-21-34814R1

Dear Dr. Klukowska-Rötzler,

We’re pleased to inform you that your manuscript has been judged scientifically suitable for publication and will be formally accepted for publication once it meets all outstanding technical requirements.

Kind regards,

Hans-Peter Simmen, M.D., Professor of Surgery

Academic Editor

PLOS ONE
---

## [Editor Report · Acceptance letter]

30 Jun 2022

PONE-D-21-34814R1 

Injury patterns of non-fatal accidents related to ice hockey, an analysis of 7 years of admission to a Level-1 Emergency Centre in Switzerland 

Dear Dr. Klukowska-Rötzler:

I'm pleased to inform you that your manuscript has been deemed suitable for publication in PLOS ONE. Congratulations! Your manuscript is now with our production department. 

Kind regards, 

on behalf of

Dr. Hans-Peter Simmen 

Academic Editor

PLOS ONE